# “Be the Dream Queen”: Gender Performativity, Femininity, and Transgender Sex Workers in China

**DOI:** 10.3390/ijerph182111168

**Published:** 2021-10-24

**Authors:** Eileen Y. H. Tsang

**Affiliations:** Department of Social and Behavioral Sciences, City University of Hong Kong, Hong Kong 852, China; eileen@cityu.edu.hk

**Keywords:** transgender sex workers, femininity, performativity, clients, intimate sex partners, hate crime

## Abstract

An under-researched aspect of transgender sex workers in China pertains to their desires and expressions of femininity. Male-to-Female (MTF) transgender sex workers are a high-risk population prone to depression and stress regarding body image, intimate relationships marked by violence, and social stigma, rendering them vulnerable to hate crimes and discrimination. Ethnographic data from in-depth interviews with 49 MTF transgender sex workers indicate that sex, gender and feminine desire are mutable in the construction of self and subjectivity. This study uses the conceptual framework of gender performativity, that is, gender is performative and distinct from physical bodies and binary classifications. It is not only an individual’s normative gender expressions which are based on the sex assigned at birth, but it also reinforces the normative gender performances of the gender binary. This article argues that the 49 MTF transgender sex workers are embodiments of gendered performances, displaying femininity to ameliorate hate crimes and discrimination as well as reinforce the masculinity and sexuality of their clients and intimate sex partners. Embracing their femininity constitutes a self-help program, enabling them to build self-confidence and develop a positive self-image in the face of overwhelming social disapproval.

## 1. Introduction

The author first met Lily (27), a male-to-female (MTF) transgender sex worker (hereafter, TSW/TSWs) in summer 2017, at her stage show in a high-end bar in Tianjin. She is tall and slender, with alluring make up and costumes. Her performance demands she wears dramatic makeup to match the five-inch stiletto heels, long voluminous hair, and tight spandex clothing. She commands the stage, alternating between vigorous up-tempo dance tunes, and slow ballads, which allow her to gaze soulfully into the eyes of her nearest admirer. Blending an array of Chinese pop tunes and Western stars such as Lady GaGa, she shows her versatility with some Chinese opera, complete with heavy makeup and traditional Chinese evening gowns. She says,

Being feminine makes me happy and is the right thing for me as a transgender woman. In my bar, I can perform a ballad by Lady Gaga, Madonna, or another celebrity you may be familiar with. I will sing and even dance like them. I can wiggle my hips a little bit, and it’s fine. You can also transform yourself to do Chinese opera with the beautiful gowns and makeup. I am not a drag queen, a man with a penis. I am a transgender queen, a man with artificial breasts and penis. I have a woman’s breasts, but I also keep my penis as my key selling point for those heterosexual and bisexual men! My clients thought they would touch a pussy, but it was a cock. My clients were excited, and they don’t need to hide and act, just be themselves, be a bisexual man!

The conversation with Lily (27) highlights some major themes regarding how Chinese TSWs capitalize their femininity. The 49 individuals interviewed were assigned male sex at birth, but their feminine identity emerged over time [1]. Those who identify outside of the gender binary are often referred to as “non-binary”. Non-binary gender identity can reflect identifying as neither male nor female, both male and female, or manifesting different genders at different times [2]. The term “transgender” fits in a non-binary gender category [2], which may combine traits historically described in terms of “male” or “female”. Transgender is an umbrella term for individuals whose gender identity or expression is different from the cultural and social expectations placed upon them based upon the sex assigned to them at birth. The TSWs interviewed provide a case for highlighting the temporal aspect of gender performance and femininity, in which their commitment toward commercial sex indicates a positive outcome in their gender affirmation that does not necessarily lead to full gender confirmation surgery.

The conventional literature involving relationships between clients and sex workers has typically examined female sex workers and heterosexual male clients [3,4]. The literature on TSWs tends to focus on negative aspects, such as portraying them as victims of discrimination and violence [5,6,7,8,9,10,11] or victims of HIV and STDs [12,13,14,15]. Thus, the literature calls for studies to examine how TSWs deal with stigma and discrimination as well as engage in relationships with their clients—and perhaps, more importantly—with their intimate sex partners [16,17,18,19].

Drawing on the concept of performativity from Butler [5], this article argues gender is both a performance and social construction. It focuses on an under-described aspect of transgenderism, that is, sexuality and sexual practices of trans bodies such as desires and expressions [5,20,21,22,23,24,25]. For example, in his study of TSWs in the context of Brazilian society, Kulick [17] argues that TSWs generally make themselves feel sexy and attractive. It is one of the only contexts where they can experience themselves as tantalizing objects of desire and develop a sense of personal worth, self-confidence, and self-esteem.

This article reports the conclusions from three years of ethnographic study in North China, where a total of 49 TSWs were extensively interviewed. This study was driven by three primary research questions: (1) How did these TSWs develop their feminine selves? (2) What is the space occupied by these TSWs who present themselves as female in every way except for the male sex organ, which is offered to clients and intimate sex partners? (3) In what ways do this cohort of TSWs attract clients and reinforce their masculinity through feminine sexuality? These findings contribute to the empirical and theoretical understanding of how femininity is constructed and performed by MTF transgender prostitution activities in a non-Western context.

## 2. Conceptual Framework

The conventional literature about transgender individuals often portrays them as marginalized socially, economically, politically, and legally [10]. Transgender people face a higher risk of mental health issues due to stigma or discrimination than general populations and even non-transgender sexual minority populations [26]. Significant numbers of transgender people in Western countries report high levels of depression, ranging from 35% to 62% [27]; anxiety, ranging from 34.5% to 40.4% [28]; and attempted suicide, ranging from 16% to 46% [29].

In Southeast Asia, literature reported the positive experiences of transgender women in the Philippines who migrated to Japan and became entertainers [30,31]; Canoy [32] studied the intersectionality of intimacy among gay and transgender identities; David [33] studied transgender labor; and Inton [34] investigated the representations of *the bakla* (the term used for both gay men and transgender woman) as male homosexual and transgender identities in Philippine cinema.

In China, depression is prevalent among a significant number of transgender people (43.4%) and a majority of those reporting depression admitted multiple attempts at suicide (60.5%) [28]. Family rejection and abandonment, and discrimination are also significant factors [35]. More study is needed regarding body image and how MTF transgender individuals view themselves. For example, how do TSWs use technologies of embodiment to create their feminine self and adapt to the expectations of heterosexual and bisexual clients as well as their intimate sex partners?

## 3. Holistic Approach

Femininity and identity are socially constructed for self-comfort [36]. As such, TSWs may capitalize technologies of embodiment to enhance their relationships with clients and intimate sex partners. TSWs occupy a sexual space that is self-comforting, and helps clients and partners reassert their masculinity and sexuality.

### 3.1. Femininity and Identity Are Socially Constructed for Self-Comfort

Gender performativity posits that identity is an outcome of repetitive performances, which constitute a particular sex, gender, or sexual identity, thus shaping the individual who enacts these performances [36,37]. Gender performativity is distinct from physical bodies and binary classifications [38]. It is not only an individual’s normative gender expressions that are based on the sex assigned at birth, but it also reinforces the normative gender performances of the gender binary. Butler [37] stated there is “no gender identity behind the expressions of gender that identity is performatively constituted by the very ‘expression’ that are said to be its results”. The notion of femininity performed by transgender women is inscribed on bodies but not on people designated female at birth [18]. Gender roles and expectations surrounding gender expression are partly, or mostly, socially constructed and reconstructed through iterations of gender “performance” and finding a place where the individual feels comfortable with their own self [26].

Western patriarchal societies use heteronormative gender ideals as a tool to simplify complex identities. Butler [36] asserted hegemonic heterosexuality portrays itself as superior to homosexuality in that it presents those who fit the dominant gender as the idealized version. Butler [36] noted that most people fail to meet the expectations of heterosexuality because not everyone fits gender stereotypes. Gender norms are subverted when they are repeated in a parodied fashion or in a context that defies expectation. Resistance and change can happen through performing bad or “faulty versions” of gendered identities. As such, transgender people might be disparaged as performing “faulty” versions of gendered identities [39], and these so-called inappropriate gendered identities have the potential to be subversive; for example, a transgender woman undergoing artificial breast surgery but decides to keep their penis. Specifically, Butler’s theory of performativity suggests that society wields influence upon the body itself, both directly and indirectly. Performativity provides a platform for the endless combinations and expressions of sex/gender/sexuality [37]. Performativity in this regard is the means by which identity is reinforced and affirmed.

### 3.2. Capitalize Technologies of Embodiment to Enhance Femininity

The pursuit of beauty allows transgender women “to construct themselves as social subjects who feel powerful and desired” [40]. For transgender women to achieve femininity, there is a close relationship between technologies of embodiment and their bodies for the development of the self, resulting in enhanced self-esteem and self-confidence. Technologies of embodiment refer to the processes by which transgender women produce, transform or manipulate their bodies through particular kinds of body-related work that signify divergent images of beauty and body figures [41]. Like other technologies, those of embodiment evolve rapidly, are quickly consumed and can swiftly respond to evolving standards of beauty to instantly reshape the user. Specific technological cosmetic procedures such as breast and eye enlargement, skin lightening and darkening, Botox treatments and nose reconstruction can optimize physical appearance and induce clients to feel aroused [42,43,44]. In the sex industry, these technologies function as tools to allow workers to manipulate their bodies or alter their embodied performance of femininity as they interact with the outside world. Johnson [45] analyzed the ‘glamour’ and ‘style’ displayed by gay entries in local beauty contests in the southern Philippines. Ochoa’s [18] concept of “spectacular femininities” invokes “a kind of hyper femininity intended for display, [...] created as the objects of an imagined masculine gaze”. Winter and Udomsak [46] said that transgender women “displayed actual self-concepts that were strongly female-stereotyped (that is, consistent with their own beliefs about femaleness); their ideal self-concepts, and aspirations for change were distinctly less female-stereotyped”.

### 3.3. Reciprocal Relationship with Heterosexual and Bisexual Clients and Intimate Partners

The feminine self that is constructed by the TSWs benefits them as well as the clients and intimate partners who construct masculinity and sexuality. The transgender women selling commercial sex by capitalizing their femininity present possibilities for pushing the boundaries of heterosexuality [47], offering an opportunity for resisting and queering the strict boundaries of normative heterosexuality and heteronormativity. Most of the clients who paid for the sexual encounters were heterosexual and bisexual men. Their interest underscores those men’s desire for breasts and feminine features, but also their interest in enjoying the transgender women’s penis [39,48].

A man’s gender and sexuality are generally not defined by his anatomy, but rather, by what he does with that anatomy. “Soft” femininity combined with the “hard” masculinity within one body is what defines their bisexuality [39]. Heterosexual and bisexual men pay for sexual encounters with the TSWs, which provides them an opportunity to have their non-heteronormative sexual and emotional needs met. The boundedness of the paid sexual encounter allowed these men to compartmentalize their sexual identities as bisexual men but not necessarily to fit into a heteronormative gender role.

## 4. Research Method

The fieldwork took place in Tianjin, North China. It is estimated that China has 4 million individuals who are transgender—about 1 in every 350 people [49]. Transgender people struggle against a constellation of daily concerns. The risk of sexually transmitted illnesses and HIV [50], ongoing concerns with police intimidation, and the real danger of being the victim of a hate crime confirms the prevailing social stigma against transgender people [49], rendering them vulnerable mentally, physically, and emotionally [38].

Ethnographic field notes were obtained through several excursions to North China over a three-year period, from May 2016 to August 2019. During winter breaks and summer months, the author worked in a bar employing several TSWs. The time spent there was roughly three to four months per year, totaling nine to ten months of field work data collection. The trips ranged from relatively short (one week) to long (four months), comprising seven to eight trips per year to the sites of interest. This approach was better suited to academic employment obligations and more feasible than extended immersive on-site fieldwork. The fieldwork data collected should be sufficiently ‘thick’ to enable analysis of the complex web of client–worker relationships. The TSWs knew that the author was a full-time researcher but not a full-time bartender; therefore, they could maintain a professional relationship.

Much of the data collection came from two primary excursions, but with seven or eight trips per year to North China, using ethnography and in-depth interviews. The first excursion occurred between May 2016 and December 2018, when the author worked as a non-paid bartender in one of the high-end gay bars in Tianjin from May 2016 to August 2018. She conducted interviews with the 25 TSWs who worked in the bar. The second excursion for data collection in China gathered data through twelve LGBT non-governmental organizations (NGOs) in northeast China from September 2018 to August 2019. An NGO in Hong Kong helped with initial connections, and snowball sampling helped recruit additional respondents. After this second excursion, the author interviewed 24 TSWs. Therefore, all together, the author interviewed 49 TSWs, aged from 23 to 48 years of age. Most of the TSWs (n = 40) reported primary-school-level education and only nine received post-secondary education. All of the TSWs interviewed said they came from rural areas in the northeastern part of China. Around half—25 out of 49 TSWs—have experienced living with intimate sex partners.

Data comprised recorded interviews, in situ note-taking, and post-event field notes. Prior to any recorded interviews, all the respondents signed consent forms. Respondents were given a copy of the author’s business card and contact details and reminded that they could withdraw without prejudice at any stage in the project. To safeguard the rights of the informants, the author verified her credentials and was neither affiliated with the police nor the government. The informants were fully assured of confidentiality and anonymity as the author used only their current ages and assigned pseudonyms. Personal information such as official identification numbers or date of birth was not collected. Protecting informants’ privacy was of paramount concern due to the sensitive nature of the data.

Criteria for the selection of interview participants were that they self-identified as TSWs and of adult age. The interviews were conducted one-on-one, in person, in private locations. Interviews were generally one hour in length. Topics included why and how the informants became TSWs, how they connected with their sex partners and clients, and some of the main factors they considered for undergoing partial gender reassignment surgery. For example, they all opted to have female breasts, yet also decided to keep the penis. What role does this play in attracting clients? What are their images of femininity? How does the risk of hate crimes impact their mental health? How do they construct gender and identity? How do they help their clients or intimate sex partners construct masculinity?

All the interviews were recorded, transcribed, and analyzed with the guidance of grounded theory [51]. Grounded theory uses an inductive approach to analyze data collected via qualitative methods. Rather than deducing hypotheses prior to data collection, the grounded theory approach systematically applies analysis to each interview in order to generate original theory. By systematically analyzing each interview, the author developed three research themes or questions. First, how did these TSWs develop their feminine selves? Second, what is the space occupied by these TSWs who transition in every way except for retaining the male sex organ? Third, in what ways do TSWs attract clients and reinforce their masculinity through feminine sexuality? The author reviewed the data collected, re-reviewed, coded, and applied existing literature to generate a new holistic approach. She coded succinctly based on the major themes that emerged from the data collection: for example, TSWs are non-binary and it can be socially constructed; technologies of embodiment can bring confidence to the TSWs and enhance their femininity, which in turn can enhance the client’s masculinity. Therefore, the holistic conceptual framework emerges from the data. The transcripts were translated by the interviewer into English and NVivo 11.0 software was used for coding and analysis. Preliminary coding began by reading and re-reading five transcripts. A codebook was then developed to cover themes drawn from the interview guides, as well as new themes that emerged during the coding process. The ethical approval of this research protocol came from the author’s Institutional Review Board (reference number: 3-9-202003-04).

## 5. Findings and Discussion

### 5.1. Femininity for Self-Comfort to Avoid Hate Crime

A hate crime is any criminal offence which is perceived by the victim, or any other person, to be motivated by a hostility or prejudice based on protected characteristics such as disability, race, religion, and sexual orientation [52]. Acts of hate violence, such as harassment, stalking, vandalism, and physical or sexual assault, reflect more socially sanctioned expressions of transphobia, biphobia, and homophobia; all are intended to send a message to LGBTQ communities [53]. All the interviewed TSWs said they had experienced the full range of hate crimes and verbal assaults from the public and the police. Each of the 49 TSWs concluded at some point that they were suffering because they still looked too much like a man. The only way to escape the constant attacks was to go ‘all in’ through plastic surgery and cosmetic products. If others saw them as women, then perhaps the assaults would cease or at least occur less often. Jiating (24) described her hate crime experience in the neighborhood where she sold commercial sex to her clients. She says,

I wasn’t meticulous about my femininity until summer 2018 when I bumped into a local resident. She yelled at me: “Get out of this city, you are such a monster. You hurt the image of our good city, so pack your suitcase and leave our neighborhood now. We don’t want you here.” She then spit on me and ran away! I felt assaulted by hate every day because of my coarse skin, Adam’s apple, and body hair. From a distance, residents could recognize me as a man. The hate made me depressed. It affected my mental health, self-esteem, and self-confidence. Therefore, I learned that my femininity allows me to express myself and get my emotions out. For example, if I feel sad, and bump into some troublemakers or bad clients, my femininity becomes a song for everything. Afterwards, I learned that I should not live too ostentatiously in downtown Tianjin but be a dream queen!

Ailing (31) also shared her hate crime experience when she went into a female washroom. She says,

Last year I entered a female washroom and two women stared at me, then began yelling at me. Next, they used a broom, bucket, and mop to hit and punch me. Another time a resident chased after me, and cursed me: “Bitch, you look unnatural, your skin is horrible and not delicate, your makeup looks like a painted wall. Your artificial breasts are poorly made and feel so fake!” His words really hurt and made me seriously question myself. I shouted: “Do I really look like a giant shemale chimpanzee?” I resolved to completely change myself and become more feminine. To avoid such hate and discrimination, I have to change my body shape.

Yuyu (43) is a TSW who has experienced physical violence from the police when she sold sex in the nearby public park. She says,

When I stood up in the downtown public park, one resident stalked me, and had already called the police. When the police came, I paid 5000 yuan (US $750) to buy them out, but other times they sent me to a detention center. The police always accuse me of being mentally ill. They said: “Are you a shemale?” I said: “No.” The police said: “you deserved to be caught. There are so many jobs in the world, don’t you dare become a shemale? You have a neck throat (Adam’s apple)! Are you sure you are a woman? Look at you, come on, you look like a monster! With fake breasts and a penis. You make me vomit. I have to go home to wash my eyes and underwear! You are so dirty and disgusting! Get out of this city right away—go back to your hometown!

Many in the local community as well as authorities such as the police misunderstand TSWs and hold negative feelings about them [6]. The interviewed sample of TSWs shared that these expressions of hatred toward them motivated them to transition and be perceived by others as attractive authentic women. The goal was to be viewed as attractive and desirable women in all outward aspects to help shield them from hatred and discrimination, which in turn helped them improve their self-esteem and confidence.

### 5.2. Performing a Feminine Self

The TSWs in the sample reported that they all first wanted to become a woman when they were teenagers or pre-teens. They each had to find their own way to reconcile the overwhelming desire to become feminine, despite the stigma associated with rejecting one’s assigned sex [7]. All 49 TSWs said that embracing their femininity helped them maintain mental health. Many of the TSWs said they spent time in routine jobs such as factories, at construction sites, as taxi drivers, and sales clerks. They hated these jobs because co-workers misunderstood them and often rejected them. What kept them going was to stay focused on their goal of becoming sexually desirable women by using technologies of embodiment to transform their physical bodies.

Ye (24) saw herself as female since she was 10 years old,

My family forced me to play with other boys and signed me up for football (soccer) and basketball classes. I played with other boys but I literally had nothing in common with them. Boys were brutal. They teased me and were physically very aggressive. I realized that I had to make myself happy and avoid the teasing and being made fun of. I think being a woman is sexy, lovely, and adorable. I hate being a man. I had to do all the physical labor work when I was at home (countryside).

Ye eagerly sought to transition and eventually saved enough money to leave her job at a smartphone factory. After completing breast surgery, she became a TSW and says she fully utilizes her feminine body to please herself, her sex partners and clients; she says:

I am very versatile for my performance in my bar, some clients love different varieties. I can become a biker chick with a neon yellow Western hat. Go for it! I could wear an evening gown, or cardigan and skirt with heels all day long. That’s awesome too. This is how I unpack and reaffirm my identity as a woman! When I stand up on stage, I have my makeup, my hair implants. I look amazing! I am truly beautiful. It is all about finding that place where I feel comfortable. I can then not only please my clients, but myself too.

Lufeng (27) said that being transgender—to her—is a show, a drama, a talk show. It is also a look, an attitude, a personality and a temperament about life. By identifying herself as a woman, she has more confidence, and people will not stare and look down on her. She says,

I did everything for myself. I need to have fine and white skin and also tone my body through workouts to make it more attractive and seductive. I love to wear dresses, earrings, and make up…When I dress up, I become provocative and seductive, and become a young and pretty woman. I don’t want to look cheap like the poor drag queens who wear heavy makeup. I watch some videos from Taobao, Baidu, WeChat to learn more about make up, and attended cosmetic tutorial school to learn etiquette, make up, and how to make myself attractive and desirable. I am very versatile in my persona.

The 49 participants consciously strive to have good skin and hair, dress, and act to embody the social expectations and perceptions of how a “beautiful woman” should appear. Each of the interviewees believe that their identity and ultimate success are inextricably linked to their body’s appearance. These TSWs’ internal notion of the self is revealed and constituted as feminine [54,55,56]. It is the embodied situations for the link between bodies and subjectivities. They capitalize femininity as a means of searching for a unique understanding of the self. Beauty is a process of empowerment that transgender people engage in so as to make sense of their identities and position themselves in the world.

Xiaomiu (27) has been working in commercial sex for one year and believes white skin can cover her ugliness. She says,

People in urban China prefer people to have lighter skin rather than darker. The slogan is “white makes you win and covers ugliness.” Therefore, I have to work harder to lighten my skin so I don’t look like a country bumpkin. I know my clients love Korean or Japanese girls as well. They love the “little flesh meat” style of TSWs. I will follow their lead to whiten my skin…

Liho (26) says a balanced diet can help her body shape. She is very meticulous to her physical appearance. She says,

I went to a dermatologist for Botox shots and regularly make trips to the gym. I inject with whitening needles twice a week. I also had to use chilled tea-bags, cucumber, potato slices, and some quality name brand whitening creams. I added antioxidant foods to my diet like berries, cherries, black plums, kidney beans, and prunes. I also reduced my alcohol intake, quit smoking and even stopped taking drugs. My whitening tips are applying 3–4 tiny drops of organic cold pressed rosehip oil after cleansing my face before bed. I also added a few drops to sunscreen and moisturizer in the morning to avoid blackspots. I was very irritated by the procedure, but each day, I could see my skin was improving. It was encouraging and kept me motivated.

Meiyi (32) says,

I take good care of my health by adapting a rigorous regimen of exercise, grooming, and skin care to fight obvious signs of aging. I even use face lifts, plastic surgery, and breast augmentation. I wear age-inappropriate clothing to appear younger. However, in a highly competitive job like being a TSW, my age (aged 32) puts me at a disadvantage. If a client wants me to have tan skin, I use makeup to make myself look like I come from the Bahamas, Costa Rica, or some other country. If another client wants me with fine-textured delicate light skin, I can go to a beauty parlor to get that too.

The 49 TSWs relied on regular exercise to make their body trim and strong. Peiwei (35) says,

I went to a gym with a no-nonsense approach for 3 months. I also need to be on a strict diet. I had to eat high protein food like steamed chicken breasts, white fish, and lots of vegetables. I was not allowed to eat carbohydrates food like noodles and rice, and it really killed me. I quit alcohol, cigarettes and drugs. With my age, the gym killed me at the beginning. I worked hard on my slow metabolism, and I tried to lose more weight! My diet also includes green vegetables with avocados, almonds and olive oil added to some meals. My coach at gym room also instructs me to cook at home. I am a bad cook, and what I can do is use a rice cooker and put everything in it. Most of my friends know that in the past I have literally almost started fires in the kitchen...

More than half of the informants work with athletic trainers to exercise and maintain their toned and slim figure. Coaches design the workout routines as well as advise them on diet, caloric intake, and skin care. According to the TSWs, femininity that embodies cultural norms of beauty is the most frequently mentioned characteristics for them. Phrases offered include “slender and slim bodies”, “having the body of a model”, “having a feminine face”, “being super feminine”, and “beauty in the form of desire”. TSWs highlight body work and body capital to cultivate desirable East Asian femininity [57] in order to cater to Chinese clients and sex partners. The benefits of these efforts are self-confidence, which aids their mental health; optimal feminine appearance, which helps them to be accepted in society and avoid hate crimes; and optimal sex appeal, which enhances their clients’ masculinity [46]. This femininity allows TSWs to develop their self-confidence as they could not rely on traditional institutions such as government or labor market groups to help them get a job. Therefore, the TSWs have to rely on themselves to reach their goals of becoming an individualized self with freedom and autonomy.

### 5.3. Performing Femininity According to Male Clients’ Gaze

The TSWs transition through sex reassignment surgery using hormone injection, thyroplasty, and breast reconstruction surgery. Only the penis is preserved. The body in this regard serves as an active agent endowed with the capacity to participate in the making of gender and the creation of social meaning of femininity [58].

### 5.4. Hormone Injections

Hormone injections are an integral part of the process of transitioning. All the TSWs interviewed admitted undergoing plastic surgery and using endogenous hormones such as estrogen blockers, which decrease libido and the size of the testicles. Some physical changes brought by estrogen may include softer skin, larger breasts, slower hair growth and decreased muscle mass. Transgender women wanting breast augmentation surgery are typically recommended to go through 12 months of feminizing hormone therapy to maximize breast growth and obtain better surgical results [59]. All the TSWs interviewed completed regimens of hormone injections.

Songlin (35) said she needed hormone injections as well as silicone injections to provide a more feminine body shape,

In addition, I have hair implants and had my jawbone adjusted to make my face look smaller. I removed my body hair as I don’t want to be a bearded queen. Clients would feel disgusted if they saw my body hair! I had lip surgery to make myself sexy as well.

Gender dysphoria refers to the state of generalized unhappiness, frustration, restlessness, and depression that occurs because of incongruence between one’s sex assigned at birth and one’s gender identity. Zihan (31) is a TSW who struggled with her self-image and says,

Dysphoria’s an evil, and every time I stood in front of the mirror, I wanted to break it into a million pieces. I had to inject hormones twice per week to get rid of my male sexual traits. Also, my voice is somewhat loud and masculine, not gentle and feminine. It is not ladylike at all. When I talk, sometimes people laugh and stare at me. I feel very embarrassed. I went to Thailand for throat surgery [thyroplasty] which injects fat into my vocal cords to make my voice soft and gentle ….

### 5.5. Gender Reassignment Surgery

All the participating TSWs feel that having breasts as well as a penis are the most authentic gender embodiment of a TSW. None of the TSWs underwent vagina surgery (vaginoplasty) and many of them asserted they would not even consider it. One of their toughest challenges is getting breast reconstruction surgery without the consent of parents. Twenty out of 28 TSWs said they went to a private hospital or approached an underground doctor to avoid filling out consent forms that required their parent’s permission [60]. It also helped them avoid continuous psychological assessments and scrutinization from doctors, counsellors, and psychologists. Eight of the study’s TSWs went to Thailand for breast reconstruction surgery. Jada (29) did her breast reconstruction surgery in Thailand in 2017.

E:Why do those heterosexual men come looking for you?

Jada:They love my artificial boobs. I got huge artificial boobs from Thailand which are made of silicone. Actually, I think they look natural, feel very authentic.

E:How could you cross the border to come back to China?

Jada:The boobs were detachable. I put it in my suitcase with a sealed ice bag. The artificial breasts could make things easier with clients as this is what they like. My breasts implants can hold for 10 years… Probably, when the breasts reach the day of expiry, I would be too old to stay in the commercial sex business.

E:How did the clients react after seeing your penis but with your breasts?

Jada:Oh my god! They loved it! None of the gay guys come to me; they hate sissy women. Most of my clients are married heterosexual men and bisexual men. They have children and wives. They come because of curiosity. Heterosexual and bisexual guys are excited and surprised to see how I have a woman’s body with a genuine authentic penis. They don’t like artificial vagina at all. Their curiosity arouses them and gives them all these fantasy ideas.

In order to remain competitive in China’s sex trade industry, TSWs use their bodies as spaces of negotiation to please their clients and resist their socially and economically vulnerable situations [37,61]. Both gender identification and the desires of male clients for TSWs are neither binary nor based on biological sex. On the contrary, femininity is constructed fluidly according to the tastes of the male clients [19]. The 49 participants chose to enter the sex industry because they realized the unique combination of a woman’s breasts and a man’s penis would attract a niche market of male clients in China. The responses of the participants to male client fantasies via their “unnatural” body create new gender identities as transgender women. It allows the TSWs to take advantage of the economic opportunities created by the market to produce and sell their transgender femininities in the sex trade industry. When clients are physically aggressive and display their dominant “machismo” persona, the TSWs match with an appropriate performance of submission. However, when clients prefer to mix other kinds of sexual activities, the TSWs can accommodate a range of masculine or feminine roles to satisfy the needs of their clients.

### 5.6. Reassert Man’s Masculinity and Sexuality for Their Clients and Intimate Sex Partners

Around half—25 out of 49 TSWs—have experienced living with intimate sex partners. When the author met Mia (27) in a high-end gay bar in Tianjin, she was performing on stage. The author was astounded by the florescent green lights and loud and vibrant pop music echoing through the theater in Tianjian. She had lived with her intimate sex partner for two years.

E:How did you know you help your partner in terms of masculinity and sexuality?

Mia:Of course, I know. My partner always says: “When I look at a feminine body, without a throat (Adam’s apple), with the artificial breasts, it really aroused me sexually. I am a direct man, and, when I touched your cock, I felt good. I have to admit I am not a straight guy, perhaps I should think more about my sexuality.”

E:In what ways did you think you help the masculinity of your partner and clients?

Mia:My partner is economically vulnerable, and unable to find a stable job, and he had to rely on my salary to feed him. Living with me makes him feel very comfortable with his bisexual identity. He did not feel guilty if he could not find a job or be the breadwinner to take care of me. He feels fine being a kept man. I will do whatever he asks me to do. It fulfils his masculine fantasy as a man!

Qingqing (35) said she lived with her intimate partner for 3 years. Her partner drives a taxi and financially relies on Qingqing. Most of the intimate partners of TSWs seem to be relatively vulnerable heterosexual or bisexual men. Qingqing said one day her partner bluntly said that he discovered something about himself by being with her. Qingqing says,

My partner always says: “Why do I have to force myself to be a heterosexual man? He is a bisexual man! I think it makes better sense for me to be bisexual. I don’t have to put up a façade and hide my sexual orientation. I can be carefree in front of you!” I guess my service created a comfort zone for him. He doesn’t need to hide and act different from who he is. Life is not a drama and he can be himself around me… save the acting and hiding his sexuality when he is in front of his parents…

The men’s desire for transgender bodies was understood to be a signifier or expression of their bisexuality for their clients or their intimate partners. Those desires mean they can be “bisexual” and reinforce their bisexuality. Men can negotiate and navigate more fluid sexual identities when they pay for a sexual encounter or cohabit with a TSW. Through the bodily experiences as a TSW, the interviewees have developed self-worth and identity as they align with their inner definition of self—that is, a beautiful, attractive woman who is able to catch the attentive male gaze [56]. They recognize their self-worth, and the world is unfair and discriminates against them. What satisfaction is derived from sex work comes from two main sources: the physical pleasure from engaging in various sexual activities with their clients, and the personal satisfaction from knowing they are expert at satisfying the needs and desires of clients. TSWs feel empowered and desired in the pursuit of beauty and performance of femininity [40,62]. The TSWs said anatomical body reconstruction enables them to live in a comfort zone where they can be feminine and enjoy their homonormativity as a bisexual man. The femininity of the TSWs affirms the clients’ identity as bisexual men, which challenges hegemonic masculinity and heteronormativity when clients solicit sex services from the TSWs. The TSWs hold a unique form of power over their clients. For example, the TSWs hold the unique power to get their clients to feel comfortable to be a bisexual man, and they did not hide their bisexuality when they bought commercial sex from them.

### 5.7. Femininity of Sexual Potency

From the numerous stories told by the 49 interviewees, it is no exaggeration to say that collectively, over a thousand clients admitted some sort of sexual potency problem. However, the TSWs insist their encounters are not solely for sex; it is not simply dallying with men who are hungry and horny. The TSWs see themselves as also providing a level of therapeutic service to address their clients’ sexual impotency. Men who see themselves as macho and masculine associate with terms such as hard, strong, and aggressive. These characteristics are the essence of male sexuality in our society [63]. However, a man who cannot maintain an erection is regarded as desexualized and emasculated [64]. Wenyi (21) says,

Some clients have erectile problems and cannot get hard; they come to find me because of my femininity and emotional work. I won’t feel unhappy, I am sure when they have sex with their girlfriends or wives, and they will feel embarrassed and lose face. Clients know I try every way I can to get them hard. I won’t tease them. Therefore, I know I provide a comfort zone for them. They are happy to stay with me. Sometimes, I help them by giving a blow job, but even then, they still cannot get hard. They seemed not to feel embarrassed in front of me though. I feel like I really help men like that find their dignity.

Haini (24) says she has developed communication skills to talk with clients who have erectile problems. She says,

I offer encouragement or therapeutic service to those clients who cannot get erect, like finding a professional counselor to heal. Most of them could improve their sexual behavior. I am proud that I know how to help those men talk about their problem and their difficulty to have sex. They admit things to me they would never say to their girlfriends, wives, or friends.

Weige (37) said she once met a heterosexual client who was married with two children. He had erectile problems for more than 10 years and could not get hard, even for one minute. She says,

This heterosexual man came looking for me, and he said: “You know what, I cannot get erect properly, and it is impossible for me to get a hard on, not even for a minute. It is so embarrassing and frustrating. I haven’t had sex with my wife for almost a year. I am so devastated I am suicidal! This is the most painful part of being a man! It is so shameful!” Therefore, I just listened to him and encouraged him to speak out his true feelings and disclose his thoughts. I am sure he would be silent in front of his wife. So I feel like I provided counseling for him. He paid me, not for sex, but for therapy.

The TSWs tend to employ more effective communication skills with emotion to talk to those men who have difficulty to perform sexually. With their femininity, good communication skills and emotional labor, the TSWs offer a comfort zone for their clients to disclose their sexual problems.

The author returned to Tianjin in December 2019 before the outbreak of COVID-19. Meeting up with Lily (27) brought me up-to-date on her life in Tianjin, and she re-affirmed that she loves her job as a TSW. Her skin was still very delicate, natural, and her hair was curly and long. She still wears her 5-inch high heels. She said: “I found myself comfortable and my life has meaning. However, I really hope society can embrace differences and accept people like me. I believe God closes a door but will open a window for me.” The author detected a spark from Lily’s eyes. Lily’s simple wish shines a light on the need for people to tolerate difference and give space for transgender people in China. Transgender people rely on self-help rather than any support from the Chinese government, and Lily hoped that transgender people would one day be treated with respect, dignity, and opportunity.

## 6. Conclusions

This article examines how subjectivities and bodies interact for the formation of identities of TSWs in China. The gender performativity provides a useful lens to rethink the issues faced by TSWs, and how they negotiate their own femininity with their clients and intimate sex partners while avoiding hate crimes, public stigma and discrimination. The 49 participants in this article have not undergone complete male to female anatomical transformation. The notion of femininity performed by TSWs is inscribed on bodies but not associated with their sex at birth. The findings contribute to the empirical and theoretical understanding of how gender and femininity is socially constructed and performed in transgender prostitution in a non-Western context. Empirically, the participants make use of technologies of embodiment such as feminine clothing, makeup, skin lightener, using cosmetic products, and hair implants to correspond to their inner notion of gender identification. They display their self-concepts of femaleness that embody cultural norms of beauty from a uniquely blended male–female perspective. The gender performativity of the TSWs is to please their male clients, but in return, their identities as transgender people are reinforced as they feel powerful and desired.

The article also demonstrates how TSWs use their bodies as spaces of negotiation and resistance against social and economic vulnerability. Using femininity to attract their clients provides a platform for understanding what the body means in a given social context, and the processes and practices that produce that body as part of the social process of becoming. Through their identity as a TSW, the participants develop a sense of personal worth and self-confidence through being desired. However, their sense of empowerment is closely linked to the satisfaction of their male clients because there are so few areas of support for their identities and profession in China. The distinct lack of systematic and consistent polices in labor law, the criminal justice system, and even needing protection from the police, all put the TSWs at risk.

Future research might expand the pool of interviewing to include TSWs who are female-to-male (FTM). In addition, much more research is needed regarding significant issues faced by all transgender people in China. This study was restricted to TSWs. The cohort interviewed for this study revealed they must look to each other for support, developing their own networks for self-help. The TSWs who were interviewed expect little from the Chinese government and even less from local law enforcement. Social change begins with awareness, and these findings offer a tentative first step.

## Data Availability

The authorship of the data owned by the author.

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
