# Peer review of "“Be the Dream Queen”: Gender Performativity, Femininity, and Transgender Sex Workers in China"

_ijerph, 2021, doi:10.3390/ijerph182111168_

Round 1

Reviewer 1 Report

Authors addressed all concerns appropriately. 

Reviewer 2 Report

The author did a nice job revising this paper to sharpen the message and clarify theory and methods for readers who my be unfamiliar or less familiar with the approach. The author also presents thorough fieldwork to illuminate the lives of MTF transgender sex workers and bring humanity to their experiences. 

This manuscript is a resubmission of an earlier submission. The following is a list of the peer review reports and author responses from that submission.

Round 1

Reviewer 1 Report

General statement:

The study argues that female transgender sex workers utilize gendered performance to avoid victimization and to reinforce gender expression of male sex partners. Authors draw the conclusion that this act of performance of being female functions as a “self-help program”. The methodology is appropriate for the research. Findings were mixed with the discussion sections and appeared to be a bit convoluted. The connection to the theoretical framework could have been stronger. Authors did not provide a limitations section or suggestions for future research.   

It is important to keep in mind who the audience of this research study will be. Words are very powerful and therefore, have the potential to provide support for members of the transgender community or perpetuate existing bias and further alienate this vulnerable population. Since the language used in the manuscript is not consistent and, at times, includes all expressions of being transgender, revisions are necessary before the manuscript is eligible for publication.

Abstract: 

The abstract includes all necessary sections but needs to be revised addressing issues of inconsistency in terminology.

Introduction:

The section would benefit from a reorganization. It is not clear why authors chose to start the section with parts of an interview with a participant. The language used is triggering and problematic painting a one-sided image of transgender individuals. The section contains unfounded facts which need to be revised. The purpose of the study is rooted in the existing literature regarding the theoretical framework used and authors provide a few of the prominent definitions. Authors need to be aware of the language used and edit to avoid implicit bias. The research question was clearly stated at the end of the section.

Methods:

The authors clearly outlined all procedural aspects in terms of data collection. This section needs some additional clarification regarding coding and data analysis. Section titled “conceptual framework” would be a better fit in the introduction section to provide a more detailed background for the study and cultural context. Authors did not provide a context for the section “holistic approach”. Based on the content, this section seems to fit in the introduction section. The methods section would benefit from a more careful examination of how the framework informed coding procedures. How was the holistic approach employed by the researchers? What was the purpose of this section.  Section Capitalize technology of embodiment to enhance femininity addresses transgender women and authors do not relate this content to TSW. As mentioned above, this distinctions needs to be clear throughout that manuscript.

Findings/Discussion: 

The sections addressing the findings of this study was combined with the discussion. At times this style was convoluted and direct citations from participants were not properly marked and too long. The organization utilizing subheadings was very well executed. This section contained a lot of theory mixed with statements of over-generalizations. Those statements were marked in the in-text comments of this review. There are multiple instances where authors used transgender women interchangeably with TSW. The discussion section would further benefit from some editing for implicit bias as highlighted by the reviewer in the text document.

Conclusion

The conclusion section is concise and clearly summarized the premise of the manuscript. It would have been stronger with statements regarding potential for future research and the societal impact of the topic.

Reviewer 2 Report

The author provides a qualitive study about the subjective experiences and construction of self of 49 transgender sex workers who identify as women and elected to keep their penises. The author explores three research questions:

  1. How did these TSWs develop their feminine selves?
  2. What is the space occupied by these TSWs who remake themselves as female in every way except for the male sex organ which is offered to clients and intimate sex partners?
  3. In what ways do TSWs attract clients and reinforce their masculinity through feminine sexuality?

The author argues gender as performativity and offers three areas to empirically understand this:

  1. Femininity and identity are social constructed for self-comfort
  2. Technologies of embodiment are capitalized to enhance femininity
  3. Reciprocal relationships with clients and intimate sex partners reassert their masculinity and sexuality

The author spent extensive time conducting fieldwork and paid particular care and attention to understanding the perspective of informants. This was an interesting article and this reviewer enjoyed reading it.

Suggestions about formatting and structure:

  • Throughout the paper, the phrasing “transgenders” is used (e.g., “Transgenders struggle against a constellation of daily concerns.” I wonder if it would be more appropriate language to use the phrasing “transgender people” as opposed to simply “transgenders.”

  • Research method – It may be useful to readers to understand why certain quotes by certain informants were elected by the author to share in the findings section of this paper.

  • The Conceptual Framework section may be better positioned before the Research Methods section or as part of the Introduction section.

  • Page 4, lines 164-169 seem better positioned in the Findings and Discussion section.

  • The Holistic Approach sections seem out of place, adding unnecessary length and extraneous sub-sections to the paper. Perhaps segments of this section may be better positioned in the Introduction section, interwoven with the findings to help interpret the empirical data, or presented in the Conclusion section to reinforce certain points.

  • Page 11, line 483, the informant’s name is Mia, but earlier in this section it is implied that the informant’s name is Nini. Some clarity can be useful here if Nini and Mia are the same or different people.

  • Regarding concision, it may not be necessary to introduce informants, individually, as a TSW if that was part of the inclusion criteria. For instance, page 10, line 425 reads, “Zihan (31) is a TSW and she says,”. By this point, the readers should know all the informants of this study were TSWs.
